# Differential Alterations in Resting State Functional Connectivity Associated with Depressive Symptoms and Early Life Adversity

**DOI:** 10.3390/brainsci11050591

**Published:** 2021-05-02

**Authors:** Eleonora Fadel, Heinz Boeker, Matti Gaertner, Andre Richter, Birgit Kleim, Erich Seifritz, Simone Grimm, Laura M. Wade-Bohleber

**Affiliations:** 1Department of Psychiatry, Psychotherapy and Psychosomatics, University Hospital of Psychiatry Zurich, 8032 Zurich, Switzerland; heinz.boeker@bli.uzh.ch (H.B.); birgit.kleim@pukzh.ch (B.K.); erich.seifritz@bli.uzh.ch (E.S.); simone.grimm@bli.uzh.ch (S.G.); laura.wade-bohleber@uzh.ch (L.M.W.-B.); 2Medical School Berlin, 14197 Berlin, Germany; matti.gaertner@medicalschool-berlin.de; 3Department of Psychiatry, Charité, Campus Benjamin Franklin, 12203 Berlin, Germany; 4Department of Consultation-Liaison-Psychiatry and Psychosomatic Medicine, University Hospital Zurich, University of Zurich, 8006 Zurich, Switzerland; andre.richter@access.uzh.ch; 5Division of Experimental Psychopathology and Psychotherapy, Department of Psychology, University of Zurich, 8050 Zurich, Switzerland; 6Psychological Institute, Zurich University of Applied Sciences, 8037 Zurich, Switzerland

**Keywords:** depression, early life adversity, functional connectivity, resting state fMRI

## Abstract

Depression and early life adversity (ELA) are associated with aberrant resting state functional connectivity (FC) of the default mode (DMN), salience (SN), and central executive networks (CEN). However, the specific and differential associations of depression and ELA with FC of these networks remain unclear. Applying a dimensional approach, here we analyzed associations of FC between major nodes of the DMN, SN, and CEN with severity of depressive symptoms and ELA defined as childhood abuse and neglect in a sample of 83 healthy and depressed subjects. Depressive symptoms were linked to increased FC within the SN and decreased FC of the SN with the DMN and CEN. Childhood abuse was associated with increased FC within the SN, whereas childhood neglect was associated with decreased FC within the SN and increased FC between the SN and the DMN. Our study thus provides evidence for differential associations of depressive symptoms and ELA with resting state FC and contributes to a clarification of previously contradictory findings. Specific FC abnormalities may underlie specific cognitive and emotional impairments. Future research should link specific clinical symptoms resulting from ELA to FC patterns thereby characterizing depression subtypes with specific neurobiological signatures.

## 1. Introduction

Depression is a common and debilitating mental disorder and constitutes one of the leading causes of disability worldwide, with a lifetime prevalence of the disease averaging 11.3% in European countries [1]. It can be understood as a complex neurobiological condition and the underlying pathophysiology is still poorly understood [2]. The overall effectiveness of available depression treatments is limited, a fact mirrored by high rates of relapse of the disease [3]. Both high prevalence rates and unsatisfying treatment outcomes have led to intense research efforts during the last decades to understand the neurobiological pathways underlying depression and optimize its treatment [4].

An extensive body of neuroimaging research now demonstrates that depression is a multifaceted syndrome linked to abnormal activity and structure of various brain regions [5]. Measures of brain activity obtained during the “resting state” (a condition where people rest in the scanner without concentrating on any task) can also be used to characterize brain network activity in depression [6]. Among the primarily studied large scale networks in depression are the default mode network (DMN), central executive network (CEN), and salience network (SN) [7]. The main nodes of the DMN are the medial prefrontal cortex, posterior cingulate/precuneus, and the inferior parietal lobule. This network is important for diverse types of affective and cognitive processes involving the self [8]. The CEN is mainly formed by the dorsolateral prefrontal and posterior parietal cortices and underpins higher executive and cognitive functioning [9]. The SN’s main nodes are the insular cortex and anterior cingulate cortex and this network is crucial for the detection of salient information as well as for switching between the DMN and CEN [10,11]. Alterations related to depression have been reported regarding within- as well as between-network functional connectivity (FC) of these networks, although findings are heterogenous, encompassing many different brain regions and with directionality of effects differing between studies [12]. One of the most consistent observations is increased FC within the DMN in depression, an observation which has been linked to rumination symptoms and excessive self-focus [13,14]. Another consistent observation is hypoconnectivity within the CEN, which may lead to cognitive impairments commonly reported in depression [15,16]. Altered FC between the DMN and CEN may underlie difficulties in switching from a default-state with attention directed internally to an executive state in which attention is allocated to external stimuli [17,18,19]. Impaired FC between neural systems involved in cognitive control and those that support salience or emotion processing may relate to deficits regulating mood [16,20].

It has been proposed that depressed individuals with experiences of early life adversity (ELA) form an etiological subtype of the disorder with specific neural signatures [21]. Not only is the exposure to ELA in the form of childhood abuse and/or neglect associated with increased vulnerability for the development of depression, but patients with such a history show a graver clinical course in terms of symptom severity and age of onset [22]. Further, this subpopulation responds more poorly to treatment in form of psychotherapy as well as pharmacotherapy [22]. ELA has been hypothesized to influence cognitive and emotional processes by disrupting the maturation of underlying brain networks [23]. Evidence so far suggests that ELA has detrimental effects on neurobiological development, resulting in functional alterations of an emotion regulation circuitry, possibly contributing to increased vulnerability for the development of depression [24].

Recently, it has been suggested that ELA leads to specific alterations in several resting state networks which also show alterations in depression. An association between decreased FC within a fronto-parietal attention network and childhood abuse has been reported [25], whereas another study observed increased connectivity within the SN as well as reduced FC between the SN and the DMN in traumatized children [26], with the latter finding being replicated in a sample of healthy adults with childhood trauma [27]. Regarding the DMN, several studies reported reduced FC within this network in association with ELA and it has been hypothesized that ELA interferes with the development of the DMN [28]. ELA has also been reported to affect FC in depressed subjects. Experiences of emotional abuse and neglect are associated with decreased FC of networks mediating attention and the processing of sensorimotor and visual stimuli in a depressed sample, linking patients’ experiences of ELA with specific functional brain network abnormalities that suggest a possible environmental contributor to neurobiological clinical symptom profiles [18]. Individuals with depression and a history of childhood neglect show more widespread FC decreases in a prefrontal-limbic-cerebellar circuitry compared to depressed individuals without such a history [29]. Further evidence underlines the necessity of not only disentangling the effects of ELA from those of depression itself but also to examine possible moderation effects, with such effects being reported regarding measures of brain activity and volume [30,31,32].

So far, moderation effects of ELA on associations between depression and FC of the DMN, SN, and CEN are still poorly understood. Furthermore, ELA may have confounded results reported by prior studies analyzing depressive samples [33]. Our study, therefore, aims to shed light on the specific alterations of DMN, CEN, and SN FC associated with depressive symptoms and ELA by analyzing a sample of healthy and depressed subjects. Different approaches can be applied when analyzing the effects of ELA. The cumulative risk model accounts for the observation that different types of adversities show high co-occurrence, focusing on the number of distinct adverse experiences rather than severity and type of experience [34,35]. Although this approach has proven useful in establishing a dose-response relationship between ELA exposure and negative health outcomes, it implies that different forms of adversity share the same underlying mechanisms, therefore showing significant limitations when used to identify mechanisms linking ELA subtypes and health outcomes [35]. Recently, an alternative model has been proposed, namely the dimensional model of adversity and psychopathology (DMAP [36,37]). The DMAP differentiates between deprivation, defined as experiences involving an absence of expected inputs from the environment, and threat, defined as experiences involving harm or threat of harm, suggesting that the two dimensions show specific effects on neural structure and function [36]. In line with the assumptions of the DMAP, it has recently been stated that different childhood trauma subtypes go along with specific neural underpinnings, with emotional maltreatment being linked especially to abnormalities in fronto-limbic socioemotional networks, neglect showing associations with abnormalities of white matter integrity and FC in various brain networks (especially an insula-based network) and sexual abuse more strongly correlating with structural deficits (especially in the reward circuit and genitosensory cortex) as well as amygdala reactivity to certain tasks [38].

In line with the assumptions of the DMAP and based on previous findings regarding specific effects of ELA subtypes on brain measurements, we decided to analyze abuse and neglect separately by focusing on emotional and physical abuse and emotional and physical neglect. Based on previous reports of sexual abuse showing distinct patterns of stress reactivity when compared with other forms of ELA [39] and its specific influence on measurements of brain structure and reactivity [38], we decided to exclude this subscale from analyses. We employed region of interest (ROI) analyses, focusing on FC between major nodes of the DMN, the SN and the CEN, and adopted a dimensional approach to test associations between severity of depressive symptoms and FC, between childhood abuse and childhood neglect and FC as well as to test moderation effects. Based on previous findings, we assumed that (1) severity of depressive symptoms would be associated with increased FC within the DMN and SN, reduced FC within the CEN, and impaired FC between the SN, DMN and CEN. Furthermore, we assumed that (2) childhood abuse would be associated with reduced FC between the DMN and SN, reduced FC within the CEN, and increased FC within the SN. For (3) childhood neglect, we assumed it would be negatively associated with FC within and between networks, especially within the SN. We also aimed at (4) testing moderating effects of ELA subtypes on the association of depressive symptoms with FC.

## 2. Materials and Methods

### 2.1. Participants and Procedures

Depressed and healthy subjects were recruited through the distribution of flyers in Zurich, Switzerland, advertisements on the internet, a short newspaper article informing about the study, and the University of Zurich mailing list. Depressed subjects were also referred by psychiatrists and psychologists affiliated with local psychotherapeutic training institutes. Before fMRI scanning, a psychologist conducted the mini-DIPS [Diagnostisches Kurz-Interview bei psychischen Störungen], a diagnostic short interview for mental disorders [40], to assess eligibility of subjects. Overall, subjects were eligible for study participation if they were between 18 and 65 years of age, presented no history of substance abuse or psychotic disease, and characteristics conform with MRI safety regulations (e.g., no pregnancy, no metallic implants, no claustrophobia, etc.). A primary DSM-IV diagnosis of Major Depressive Disorder (MDD), confirmed with the mini-DIPS, determined inclusion for depressed subjects. Exclusion criteria for healthy subjects were any type of current psychiatric disorder and a history of a depressive disorder in the past. All subjects provided written informed consent. The ethics committee of the canton of Zurich approved the study.

Forty-seven healthy subjects were included in a first step. Five healthy subjects had to be excluded from analyses due to: a. corrupted neuroimaging data files (one), b. dropout between the clinical interview and the fMRI scanning (one), c. incomplete questionnaire data (three). Seventy-six subjects suffering from depressive symptoms interested in participating in the study were included in a first step. Twenty of these subjects were excluded after the first clinical interview because: a. another psychiatric disorder was the primary diagnosis (eight), b. a history of substance abuse (two), c. the depressive symptoms had already remitted since the first contact with the study team or were not sufficiently severe for a MDD diagnosis (four), d. dropout between the clinical interview and the fMRI scanning (five), e. metal piercings that could not be removed (one). One depressed subject had a panic attack during scanning and could not complete the experiment. Due to incomplete questionnaire data, eight depressed subjects had to be excluded from following analyses. Six depressed subjects who underwent fMRI scanning reported an antidepressant medication and were excluded from the analyses to eliminate medication as a possible confound. This resulted in a final sample of 42 healthy subjects and 41 subjects with MDD.

To assess ELA, the German version of the Childhood Trauma Questionnaire (CTQ) was used [41,42]. It is a self-report questionnaire which retrospectively assesses childhood trauma until the age of 17. It comprises 25 items, forming one general score and 5 subscales (emotional, physical, and sexual abuse as well as emotional and physical neglect) with 5 items each. The emotional and physical abuse subscales were added to form the abuse score, whereas the two neglect-related subscales (emotional and physical neglect) were added to form the neglect score. The subscales showed good reliability in our sample (Cronbach’s α_abuse_ = 0.87, Cronbach’s α_neglect_ = 0.84).

To assess the severity of depressive symptoms, we used the Beck Depression Inventory II, a self-report questionnaire assessing general depression (BDI-II [43,44]). This is composed of 21 items rated on a 4-point Likert scale, with answers relating to the prior two weeks. Higher values represent a more severe depressive symptomatology. In our study, reliability was excellent (Cronbach’s α = 0.97).

### 2.2. Data Acquisition and Analysis

#### 2.2.1. fMRI Data Acquisition

Data acquisition was performed on a Philips Intera 3T whole-body MR unit equipped with a 32-channel Philips SENSE head coil. Functional time series were acquired with a sensitivity-encoded single-shot echo-planar sequence (SENSE-sshEPI [45]). The following acquisition parameters were used in the fMRI protocol: echo time = 35 ms, field of view = 220 mm × 220 mm × 128 mm, acquisition matrix = 80 × 80, voxel size: 2.75 mm × 2.75 mm × 4 mm, SENSE acceleration factor R = 2.0. Using a mid-sagittal scout image, 32 contiguous axial slices were placed along the anterior-posterior commissure plane covering the entire brain with a TR of 3000 ms (θ = 82°). The first five acquisitions were discarded to eliminate the influence of T1 saturation effects. An anatomical T1-weighted structural image was also acquired (FOV = 220 × 220 × 135 mm; acquisition matrix = 224 × 187, interpolated to 224 × 224; reconstructed voxel size = 0.98 × 0.98 × 1.5 mm^3^, 180 slices). Subjects were advised to close their eyes during the scan. Scanning time was approximately 10 min.

#### 2.2.2. fMRI Data Analysis

Resting state fMRI data was analyzed using the toolbox CONN [46]. We performed preprocessing using the CONN-default pipeline for analyses in MNI-space, which includes realignment and unwarping for motion correction, slice-time correction, automatic detection of artifactual scans (ART-based scrubbing), normalization, and spatial smoothing (using an FWHM kernel of 8 mm). Denoising included single-subject linear regression analyses to remove artifacts due to movement (12 motion covariates: motion parameters plus temporal derivatives), to physiological effects (total of 10 CompCor eigenvariates: 5 each from eroded WM and CSF masks), and to artifactual scans. Finally, the resulting BOLD time series were band-pass filtered (0.008–0.09 Hz).

For the FC analyses between regions of interest (ROIs), we employed 15 ROIs of the networks of interest (DMN, SN, CEN) from the CONN toolbox, as defined by its independent component analysis of the Human Connectome Project dataset (497 subjects). These include four ROIs for the DMN (medial prefrontal cortex (mPFC), precuneus cortex (PCC), left and right lateral parietal (LP)), seven ROIs for the SN (anterior cingulate cortex (ACC), left and right anterior insula, left and right rostral prefrontal cortex (rPFC), left and right supramarginal gyrus (SMG)) and four ROIs for the CEN (left and right lateral prefrontal cortex (LPFC), left and right posterior parietal cortex (PPC)).

#### 2.2.3. Statistical Analysis

Linear regression analyses were used to explore associations of ROI-to-ROI pairwise FC with the BDI-II score, the CTQ abuse score, and the CTQ neglect score, as well as the moderation effect of the CTQ abuse and CTQ neglect scores on the association of FC with the BDI-II-score. Analyses were performed using the toolbox CONN [46]. We used the following predictors: BDI-II score, CTQ abuse score, CTQ neglect score, BDI-II*abuse, BDI-II*neglect, age, and gender. To form the interaction terms of BDI-II with CTQ abuse and BDI-II with CTQ neglect, the BDI-II score as well as the CTQ abuse and CTQ neglect scores were mean centered and then the product between BDI-II and CTQ abuse scores and between BDI-II and CTQ neglect scores was calculated. Bivariate correlations were first calculated for each subject (1st level analyses). These Pearson’s r correlation coefficients were converted to normally distributed z-scores using Fisher transformation. We then performed ROI-to-ROI FC analyses on the group level, using bivariate correlations. To isolate effects of BDI-II when testing associations between BDI-II and FC, we controlled for CTQ abuse and CTQ neglect, the interaction terms (BDI-II*abuse and BDI-II*neglect), age, and gender. To isolate the effects of CTQ abuse and CTQ neglect, we tested associations of CTQ abuse and CTQ neglect with FC separately, controlling for either one, respectively, as well as for BDI-II, the interaction of CTQ abuse and CTQ neglect with BDI-II, age, and gender. Finally, we tested associations between interaction terms (BDI-II*abuse and BDI-II*neglect) and FC separately, controlling for either one, respectively, as well as for BDI-II, CTQ abuse, CTQ neglect, age, and gender. We used seed level false discovery rate (FDR) to correct for multiple comparisons and a threshold of *p* = 0.05.

## 3. Results

Here, we report associations of ROI-to-ROI FC with BDI-II (*M* = 14.61, *SD* = 14.08, range = 0–47), CTQ abuse (*M* = 13.81, *SD* = 5.33, range = 9–46) and CTQ neglect (*M* = 15.63, *SD* = 5.87, Range = 10–36) scores. Finally, we report observed moderation effects of CTQ abuse and CTQ neglect on the association of ROI-to-ROI FC with BDI-II. Table 1 describes the sociodemographic and psychometric characteristics of the sample.

### 3.1. Associations between Functional Connectivity and BDI-II

We observed positive associations between the BDI-II score and FC within the SN (regarding following pairs of nodes: left rPFC–right rPFC, left rPFC–ACC, and left rPFC–right SMG). We found negative associations between BDI-II and FC between the SN and DMN (left rPFC–left LP, left rPFC–right LP). Furthermore, we observed negative associations between BDI-II and FC between the SN and CEN (left rPFC–left LPFC, left rPFC–left PPC, and left rPFC–right PPC). These findings are listed in Table 2 and illustrated in Figure 1 (see Appendix A in the Appendix A for associations between BDI-II and ROI-to-ROI-FC for the depressed group, the healthy group, as well as combined).

### 3.2. Associations between Functional Connectivity and CTQ Abuse and CTQ Neglect

CTQ abuse was positively associated with FC within the SN, namely with FC of the right insula with the right rPFC (see Table 3, as well as Appendix A in the Appendix A). CTQ neglect was positively associated with FC of the SN with the DMN (PCC–left SMG, PCC–right SMG, PCC–right rPFC, mPFC–right rPFC). We observed a negative association between CTQ neglect and FC within the SN, namely FC of the right rPFC with the right insula (see Table 3 and Figure 2, as well as Appendix A in the Appendix A).

### 3.3. Moderation Effects of CTQ Abuse and CTQ Neglect on the Association between BDI-II and Functional Connectivity

The interaction term BDI-II*abuse was negatively associated with FC within the SN, namely with FC of the right insula with the right rPFC (*T* (75) = −3.24, *p*-FDR = 0.02). For participants with higher scores of CTQ abuse, BDI-II was associated with decreased FC of the right insula with the right rPFC. These associations were inverse for participants with lower scores of CTQ abuse (see Appendix A in the Appendix A). We found no significant association of FC with the interaction term BDI-II*neglect (all *p*-FDRs > 0.05).

## 4. Discussion

The aim of this study was to identify specific alterations of FC within and between three major resting state brain networks (DMN, CEN, and SN) associated with severity of depressive symptoms and ELA. Importantly, we isolated associations of depressive symptoms and ELA with FC by adding respective covariates in the analyses. We also tested moderation effects of ELA on the association between depressive symptoms and FC. Severity of depressive symptoms was positively associated with FC within the SN. Moreover, severity of depressive symptoms negatively linked with FC between the SN and DMN as well as between the SN and CEN. Childhood abuse and neglect differentially linked with FC within the SN, with abuse showing a positive association and neglect showing a negative association with within-network FC. Moreover, childhood neglect was positively associated with FC between the SN and the DMN. Childhood abuse moderated the association between depressive symptoms and FC within the SN, whereas we observed no moderation effect of childhood neglect.

### 4.1. Functional Connectivity Associated with Severity of Depressive Symptoms

In line with our hypothesis, severity of depressive symptoms was positively associated with FC within the SN. Abnormal FC within the SN has been repeatedly reported in depression [18,47,48], and recent theories link dysfunction of the SN to certain depressive traits [49,50]. Increased FC within the SN may underlie emotional over-reactivity [51], and it has been hypothesized that increased FC within the SN may contribute to ruminative responses to negative mood states and life events in patients with depression [18]. Another important function that has been ascribed to the SN is the coordination of networks, especially the disengagement of the DMN and engagement of the CEN [52]. In our study, we observed a negative association between severity of depressive symptoms and FC of the SN with the DMN and the CEN. Our results thereby support previous reports suggesting a weakened role of the SN for switching the two networks [53] and fit well into the triple network model, which assumes that abnormalities of FC within one network may affect other networks [52]. Increased FC within the SN observed in our study may underlie increased efforts to regulate mood, at the expense of its role in switching between the CEN and DMN, lastly resulting in impaired flexibility when reacting to the demands of the environment. This assumption is in line with difficulties in task switching and in switching from external to internal states in depression [15,19].

Previous literature has highlighted the role of the insula in switching between the DMN and CEN [11,54], whereas in our study the above-mentioned alterations of FC associated with depressive symptoms center on the rPFC. The role of the rPFC is complex. It has been linked with prospective memory and self-referential processing [55,56]. The “gateway hypothesis” states that the rPFC supports mechanisms enabling to attend either to environmental stimuli or to self-generated or maintained representations [57]. Our results regarding the rPFC largely align with those reported by Sheline and colleagues [58]. The authors found a frontal area, termed as the dorsal nexus, to show aberrant FC with nodes of the DMN, CEN and SN in depression and hypothesized that this area “hot-wires” the three networks together, leading to various depressive symptoms. This so-called dorsal nexus overlaps with the left rPFC ROI used in our study. In line with the gateway hypothesis and findings regarding the dorsal nexus, we propose that the rPFC complements the insula in orchestrating network interactions, specifically interactions between the DMN, a network which is associated with internal mental processes, and the CEN, a network crucial for processing external inputs [8,9,10,11].

Contrary to our hypotheses, we did not find aberrant FC within the DMN, nor within the CEN. One possibility is that in previous studies medication has been a potential confound, as our study only reports findings from unmedicated participants [12,59]. A recent study argued that reduced FC within the DMN in depressed individuals compared to healthy controls may reflect medication usage rather than illness duration [60]. Further research should clarify if specific features (e.g., antidepressant medication, chronification, length of illness) of depressive participants differentially link to the FC of the DMN, a possible explanation of the heterogeneity of results regarding this network. Similar assumptions may be drawn from our finding concerning FC within the CEN. Lastly, it has been stated that research regarding depression not controlling for the effects of ELA may not have accounted for confounding effects of the latter [33], a matter which we have been able to address in our study.

### 4.2. Functional Connectivity Associated with ELA

Childhood abuse was positively associated with FC within the SN, especially with FC between the rPFC and the insula. This finding supports the results reported by Marusak, Etkin and Thomason [26] describing increased FC of the insula with regions of the SN in a sample of trauma-exposed youth. In their study, SN FC within the insula mediated the relation between childhood trauma and reward sensitivity, suggesting that enhanced salience detection, diminished sensitivity to reward and FC changes may contribute to later cognitive and affective deficits observed in individuals who have experienced childhood trauma and therefore represent a vulnerability to the development of psychiatric diseases. This notion is of particular interest when considering the positive association that we observed between depressive symptoms and SN FC. Longitudinal studies are needed to examine if increased FC within the SN, especially of the insula, following childhood abuse serves as a predisposing factor for the later development of depressive symptoms by influencing salience detection. Our findings do not only replicate previous results [26] in a bigger, adult sample, they also extend their observations by linking a specific form of traumatization, namely childhood abuse, to increased FC within the SN.

On the contrary, childhood neglect was associated with reduced FC within the SN in our study. Reduced insular connectivity has been reported when comparing depressive patients with childhood neglect to patients without such a history [29]. The insula is activated by emotionally salient information and—together with other regions of the SN—contributes to the initiation of adaptive behavior in response to challenges. Given that neglected children lack appropriate stimulation by physical and emotional input, it has been hypothesized that the absence of these cues and a lesser need to monitor and react to them may result in reduced connectivity of the insula [38]. Considering that abuse, contrary to neglect, represents increased aversive physical and emotional input, it is not surprising that abuse shows an inverse effect on SN FC. Given the role that has been attributed to the insula in salience detection, it may be assumed that increased FC of the insula within the SN following childhood abuse is a consequence of growing up in a “dangerous” environment, resulting in SN over-reactivity due to misattributions of emotional salience to mundane events [52]. Reduced FC of the insula within the SN, because of lacking external input, may go along with weakened salience detection, resulting in difficulties to initiate adaptive behavior. Additionally, in our study childhood neglect was associated with increased FC between the SN and the DMN, a finding which has not been reported previously about traumatization. These changes may constitute a bottom-up interference of self-processing regions with emotional control and salience mapping [19]. In line with the above-described possible consequences of lacking stimulation in individuals with childhood neglect, it could be argued that these individuals tend to orient themselves more strongly towards internal processes when navigating their environment. In fact, childhood neglect involves the lack of appropriate environmental stimulation or interpersonal interaction needed by the developing brain [61]. Increased FC between the SN and DMN may therefore underlie a stronger orientation towards internal stimuli because of this lack of appropriate external stimulation during important developmental phases of the brain [29].

Taken together, our findings are in line with the notion that different childhood trauma subtypes go along with specific neural underpinnings [38] and support the DMAP’s suggestion to analyze the effects of threat (here defined as abuse) and deprivation (here defined as neglect) separately, when controlling for either one, respectively [62].

### 4.3. Moderation Effects of ELA on the Association of Severity of Depressive Symptoms and FC

We observed a moderating effect of childhood abuse on the association between severity of depressive symptoms and FC within the SN. For participants with higher scores of childhood abuse, severity of depressive symptoms was negatively associated with FC between the insula and the rPFC, whereas the association was inverse for participants with lower scores of childhood abuse. It has been proposed that inverse patterns of amygdala reactivity related to presence or absence of a history of ELA may represent different subtypes of depression, namely a disinhibited and an inhibited subtype [31]. A similar assumption could be made for the here observed patterns of SN FC. Reduced FC within the SN has been linked with apathy in depressed individuals [63]. Therefore, it could be assumed that individuals with higher levels of childhood abuse experience stronger apathy when developing depressive symptoms. Differential FC patterns may not only be related to different clinical symptom profiles but may also have implications for treatment. In a recent study, it has been proposed that lower insula FC within the SN serves as an indicator for an insufficient response to antidepressant treatment [64]. Given the negative association between severity of depressive symptoms and FC within the SN that we observed in participants with higher levels of childhood abuse, this may be an explanation for the fact that individuals with a history of ELA respond more poorly to antidepressant treatment compared to individuals without such a history [22].

Finally, our observation of a moderation effect regarding childhood abuse, but not neglect, is in line with previous reports suggesting that unique mechanisms link different types of ELA with psychopathology [62,65]. According to the DMAP, experiences of threat are supposed to particularly influence brain circuits involved in emotional processing, increasing risk for psychopathology by leading to difficulties with emotion regulation [62]. Our observation of a moderating effect of abuse on the association between severity of depressive symptoms and FC within the SN is in line with this assumption, possibly linking neural changes following abuse with emotion regulation deficits and therefore depressive symptoms. The DMAP assumes that deprivation influences psychopathology by altering the development of complex cognition and associated neural substrates (e.g., the CEN) [62], an assumption our results do not support regarding depressive symptoms. It should be noted that deprivation seems to be more strongly associated with externalizing disorders [36] and it has recently been reported that deprivation is associated with a risk for externalizing problems via effects on verbal abilities [66]. A better understanding of the neurodevelopmental mechanisms through which different forms of ELA link to psychopathology are highly relevant for treatment interventions, as it could be suggested that depressed individuals with experiences of abuse may profit more strongly from therapeutic interventions specifically focusing on emotion regulation abilities. Individuals with a history of neglect may profit more strongly from interventions targeting cognitive and verbal abilities. These hypotheses need to be addressed by future research.

### 4.4. Limitations

Our study is not without limitations. First, rather than employing a group comparison of depressed to healthy participants, we opted for a dimensional approach in our statistical analyses. This enabled us to explore associations of depressive symptoms and ELA in a relatively large and diversified sample of healthy and depressed participants. However, this dimensional approach also implied that we combined two groups in which symptom severity was unequally distributed at the danger of inflating certain associations of symptoms and FC abnormality. We opted for this choice as our depressed sample was very heterogenous and comprised participants with very mild to quite severe levels of depression and healthy participants with subclinical depressive symptoms. Regression analysis also allowed us to study the differential associations of depressive symptoms and ELA with FC by controlling for the influence of the respective predictor in our regression model. However, this dimensional approach implies that we cannot draw any conclusions concerning distinctive FC abnormalities that characterize study participants with a clinical diagnosis of depression in comparison to healthy participants. Second, in our sample women are more strongly represented than men, an aspect that should be noted in the light of research pointing to sex differences in resting state FC [67,68]. Although we controlled for sex differences in our analyses, the observed results may be more representative of influences of severity of depressive symptoms and childhood trauma in female rather than male subjects. Third, certain shortcomings of our use of behavioral measures merit to be discussed. We did not test associations of specific symptom clusters (e.g., cognitive and somatic-affective subtypes based on the BDI-II or specific depression features such as rumination) with FC. Especially regarding the discussed possible associations of abnormalities in FC of the three resting state brain networks and specific cognitive and emotional impairments, a more detailed typification based on behavioral measures may be informative. Also, we used the CTQ to retrospectively assess childhood abuse and neglect. The validity of retrospective assessment of ELA has been questioned, emphasizing that a tendency of false negatives may contribute to a substantial measurement error [69]. Only prospective studies that follow participants from childhood to adulthood would allow to overcome this methodological issue when studying the effects of ELA on resting state networks in the adult brain. Fourth, our sample is relatively young (overall mean of 29.47 years). While many depressive disorders have their onset in late adolescence and early adulthood, they most often take a recurrent and chronic treatment course [70]. Our sample is likely not representative of these recurrent and chronic treatment courses. Also, while it is well documented that ELA influences mental health outcomes across the lifespan [71,72], the surely complex association of time and neurobiological correlates of ELA remains to be clarified—a matter that we were not able to address in our study. Thus, our findings need to be replicated in samples older in age or ideally in a longitudinal study design.

## 5. Conclusions

Taken together, our findings link severity of depressive symptoms, childhood abuse and childhood neglect with specific FC abnormalities that may underlie specific cognitive and emotional impairments. Furthermore, inverse associations of severity of depressive symptoms and FC depending on experiences of ELA may explain differences in disease courses and treatment response in individuals with a history of ELA compared with individuals without such a history. Our results underline the necessity to account for ELA when studying neurobiological pathways of depression. Future research that employs additional behavioral measures and longitudinal study designs is needed to study associations of FC abnormalities and specific clinical symptoms across time.

## Figures and Tables

**Figure 1 brainsci-11-00591-f001:**
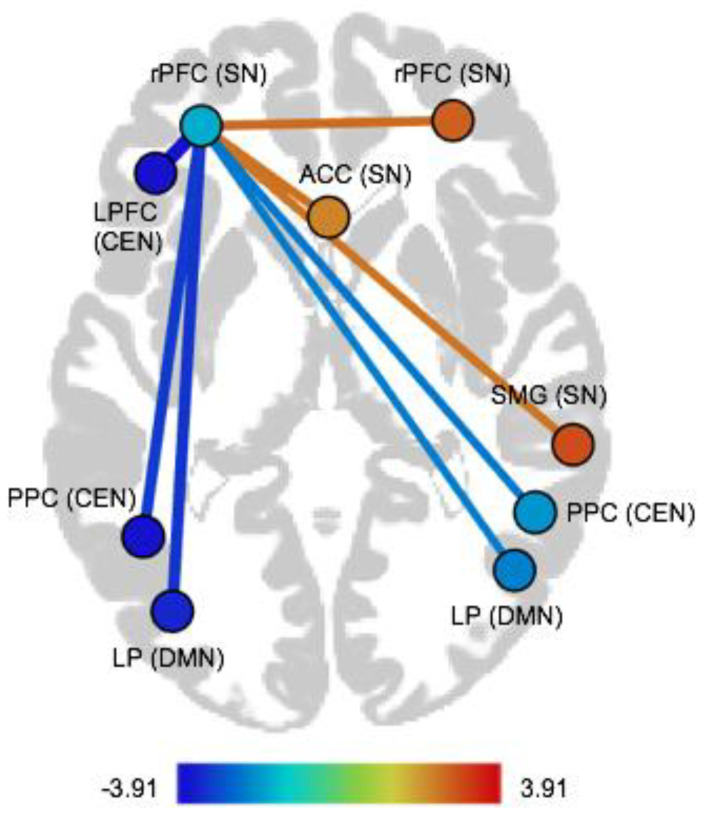
Illustration of ROI-to-ROI FC associated with the BDI-II score. rPFC = rostral prefrontal cortex, ACC = anterior cingulate cortex, LPFC = lateral prefrontal cortex, SMG = supramarginal gyrus, PPC = posterior parietal cortex, LP = lateral parietal, SN = salience network, CEN = central executive network, DMN = default mode network.

**Figure 2 brainsci-11-00591-f002:**
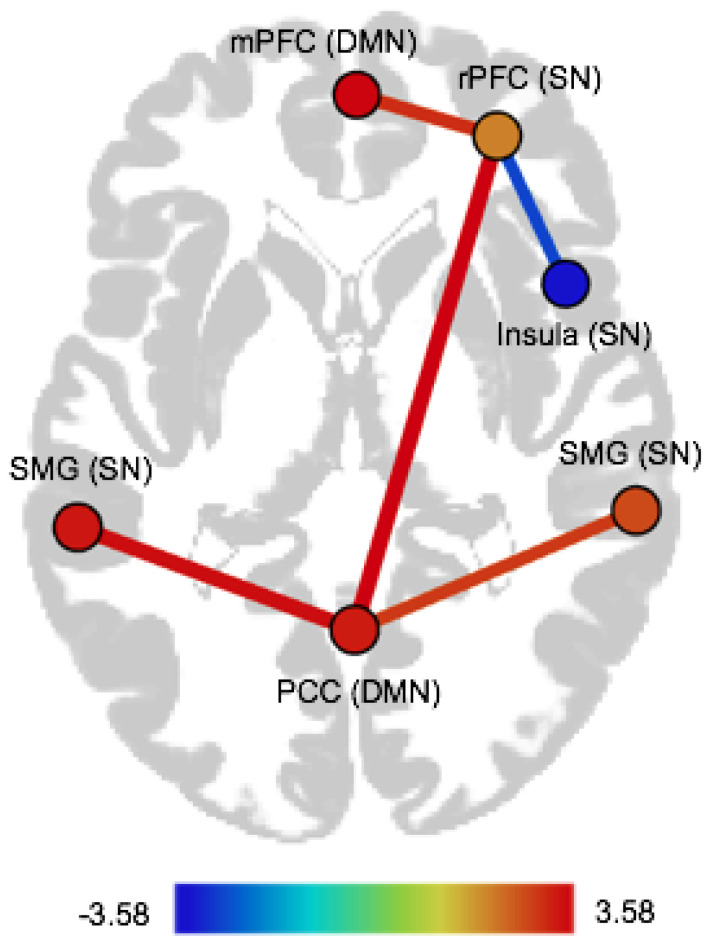
Illustration of ROI-to-ROI FC associated with CTQ neglect. mPFC = medial prefrontal cortex, rPFC = rostral prefrontal cortex, SMG = supramarginal gyrus, PCC = precuneus cortex, SN = salience network, DMN = default mode network.

**Table 1 brainsci-11-00591-t001:** Sociodemographic and psychometric characteristics of the sample.

Variables	Healthy Control Group (*n* = 42)	Depressive Patients (*n* = 41)
Age	M = 30.74 (SD = 11.22)	M = 28.2 (SD = 10.06)
Gender		
Female Male	*n* = 32 *n* = 10	*n* = 31 *n* = 10
BDI-II	M = 2.43 (SD = 2.92)	M = 27.1 (SD = 9.04)
CTQ: Neglect	M = 14.55 (SD = 5.31)	M = 16.73 (SD = 6.27)
CTQ: Abuse	M = 12.38 (SD = 3.65)	M = 15.27 (SD = 6.34)
CTQ subscales		
CTQ: SA	M = 5.12 (SD = 0.55)	M = 5.44 (SD = 1.23)
CTQ: EN	M = 8.12 (SD = 3.83)	M = 10.22 (SD = 4.53)
CTQ: PN	M = 6.43 (SD = 2.06)	M = 6.51 (SD = 2.27)
CTQ: EA	M = 7 (SD = 2.66)	M = 9 (SD = 3.80)
CTQ: PA	M = 5.38 (SD = 1.58)	M = 6.27 (SD = 3.05)

BDI-II = Beck Depression Inventory-II. CTQ = Childhood Trauma Questionnaire. CTQ: Neglect = CTQ: EN + CTQ: PN. CTQ: Abuse = CTQ: EA + CTQ: PA. SA = Sexual Abuse. EN = Emotional Neglect. PN = Physical Neglect. EA = Emotional Abuse. PA = Physical Abuse.

**Table 2 brainsci-11-00591-t002:** Significant associations between BDI-II and ROI-to-ROI FC, sorted by within- and between-network FC.

Pair of ROIs	Dir	*T*(75)	*p*-FDR
**Within-network FC**			
Salience network			
rPFC (left)–rPFC (right)	pos	2.58	0.04
rPFC (left)–ACC	pos	2.49	0.04
rPFC (left)–SMG (right)	pos	2.45	0.04
**Between-network FC**			
Salience network–Default mode network			
rPFC (left)–LP (left)	neg	−3.23	0.01
rPFC (left)–LP (right)	neg	−2.29	0.04
Salience network–Central executive network			
rPFC (left)–LPFC (left)	neg	−3.91	0.003
rPFC (left)–PPC (left)	neg	−3.12	0.01
rPFC (left)–PPC (right)	neg	−2.37	0.04

ROI = region of interest, Dir = Direction, FC = Functional connectivity, rPFC = rostral prefrontal cortex, ACC = anterior cingulate cortex, SMG = supramarginal gyrus, LP = lateral parietal, LPFC = lateral prefrontal cortex, PPC = posterior parietal cortex, pos = positive, neg = negative, FDR = false discovery rate.

**Table 3 brainsci-11-00591-t003:** Significant associations between CTQ abuse and ROI-to-ROI FC, as well as CTQ neglect and ROI-to-ROI-FC, sorted by within- and between-network FC.

Pair of ROIs	Dir	*T*(75)	*p*-FDR
**CTQ abuse**			
**Within-network FC**			
Salience network			
rPFC (right)–Insula (right)	pos	3.40	0.02
**CTQ neglect**			
**Within-network FC**			
Salience network			
rPFC (right)–Insula (right)	neg	−2.75	0.03
**Between-network FC**			
Salience network–Default mode network			
rPFC (right)–PCC	pos	3.58	0.01
rPFC (right)–mPFC	pos	3.05	0.02
SMG (left)–PCC	pos	3.44	0.01
SMG (right)–PCC	pos	2.91	0.02

ROI = region of interest, Dir = Direction, FC = Functional connectivity, rPFC = rostral prefrontal cortex, PCC = precuneus cortex, mPFC = medial prefrontal cortex, SMG = supramarginal gyrus, pos = positive, neg = negative, FDR = false discovery rate.

## Data Availability

The data presented in this study are available on request from the corresponding author. The data are not publicly available due to privacy restrictions.

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
