# Peer review of "Differential Alterations in Resting State Functional Connectivity Associated with Depressive Symptoms and Early Life Adversity"

_brainsci, 2021, doi:10.3390/brainsci11050591_

Round 1

Reviewer 1 Report

The authors have successfully addressed all my concerns and answered appropriately all my questions. The clarifications and modifications that they have made in the revised version have significantly raised the quality of their paper. I have no further requests or questions. The paper makes an excellent contribution to the field.

Author Response

We would like to thank the reviewer for the very constructive feedback and we are happy to have addressed the reviewer's points adequately.

Reviewer 2 Report

Authors need to reference Supplementary Figures in the text (only Figure S4 was referenced- Line 324). As stated in the first revision (Point #4), I would recommend at least to include CTQ neglect sig association in Table 3. I understood their rationale for excluding it but it can be misleading to readers to partly show some information in the Tables and Figures.  

Author Response

We would like to thank the reviewer for pointing out possibilites to better present our results. The suggested changes have been applied, with figures included in supplementary materials referenced in the text (see lines 348-350, 367-369 and 370-373). Furthermore, we deleted the results regarding CTQ abuse from the text and included it in table 3 instead, as suggested (see lines 375-386).

Lines 348-350: These findings are listed in table 2 and illustrated in figure 1 (see figure S1 in the supplementary materials for associations between BDI-II and ROI-to-ROI-FC for the depressed group, the healthy group, as well as combined).

Lines 367-369: CTQ abuse was positively associated with FC within the SN, namely with FC of the right insula with the right rPFC (see table 3, as well as figure S2 in the supplementary materials).

Lines 370-373: We observed a negative association between CTQ neglect and FC within the SN, namely FC of the right rPFC with the right insula (see table 3 and figure 2, as well as figure S3 in the supplementary materials).

Lines 375-386:

Table 3. Significant associations between CTQ abuse and ROI-to-ROI FC, as well as CTQ neglect and ROI-to-ROI-FC, sorted by within- and between-network FC.

Pair of ROIs

Dir

T(75)

p-FDR

CTQ abuse

Within-network FC

  Salience network

  rPFC (right) – Insula (right)

pos

3.40

.02

CTQ neglect

Within-network FC

Salience network

rPFC (right) – Insula (right)

neg

-2.75

.03

Between-network FC

Salience network – Default mode network

rPFC (right) – PCC

pos

3.58

.01

rPFC (right) - mPFC

pos

3.05

.02

SMG (left) – PCC

pos

3.44

.01

SMG (right) – PCC

pos

2.91

.02

Notes. ROI = region of interest, Dir = Direction, FC = Functional connectivity, rPFC = rostral prefrontal cortex, PCC = posterior cingulate cortex, mPFC = medial prefrontal cortex, SMG = supramarginal gyrus, pos = positive, neg = negative, unc = uncorrected, FDR = false discovery rate.

This manuscript is a resubmission of an earlier submission. The following is a list of the peer review reports and author responses from that submission.

Round 1

Reviewer 1 Report

This study examined associations between depressive symptomology, childhood abuse, and childhood neglect with resting state functional connectivity of large-scale networks (eg, DMN, CEN, SN) in a sample of 83 individuals with major depressive disorder and healthy controls. They also examined childhood abuse and neglect as a moderator of associations between depressive symptoms and resting state connectivity. The authors found that depressive symptoms were uniquely linked to increased functional connectivity of the SN, but deceased functional connectivity between the SN and DMN, as well as the SN and CEN. Childhood abuse was linked to increased SN functional connectivity, but childhood neglect was linked to decreased SN functional connectivity and increased functional connectivity between the SN and DMN. With regard to the moderator analyses, for those with higher levels of childhood abuse, depressive symptoms were associated with decreased right insula-right PFC. However, those with lower levels of childhood abuse showed the opposite pattern. While these findings are interesting and incorporated proper covariates to isolate depression versus abuse versus neglects’ impact on resting state functional connectivity, all of the statistics involving BDI scores might be unduly influenced by having a bimodal distribution, since the authors looked at BDI-II scores across the entire sample that included both healthy controls and those with major depressive disorder. The authors need to demonstrate that the BDI-II scores do not have a bimodal distribution across the full sample. If the BDI scores are bimodally distributed, then one needs to analyze the BDI-II-resting state functional connectivity associations only amongst the depressed group or the depressed group and healthy control group separately. It is also important to show scatter plots of all the associations…which could also help determine whether the BDI-II scores have a bimodal distribution. In the limitations section, I did not quite understand why a categorical approach could not be taken as well? There are CTQ subscale cut-off scores that have been used in other studies. If possible, it might be useful to do a supplementary categorical analysis (perhaps in a supplemental section).  See the following citation for an example of the categorical approach: Fan et al., (2020). Childhood trauma is associated with elevated anhedonia and altered core reward circuitry in major depression patients and healthy controls. Human Brain Mapping, 1-12. Please see below for additional comments that I hope will be helpful for enhancing the manuscript.

Introduction

  1. The authors mentioned in the introduction starting on line 101:” Since different childhood trauma subtypes have differential effects on 102 regional brain structures, activity, and connectivity [34], we decided to analyse abuse and 103 neglect separately by focusing on emotional and physical abuse and emotional and physical neglect, whereas we did not analyse the effects of sexual abuse.” It would be helpful to expand on these prior findings…what do we know about the different neural underpinnings of different trauma types? This will help make it clearer how you derived your hypotheses.
  2. Also, there is a type in the above sentences “analyse” should be “analyze”. I think “analyse” is the UK spelling.
  3. In the introduction, it was stated that the effects of sexual abuse was not analyzed. Why is that? No explanation is provided.

Methods

  1. It sounds like in the methods section that the authors looked at associations between BDI-II scores and functional connectivity across healthy controls and those with major depressive disorder? If so, this is likely highly problematic, given that you likely have a bimodal distribution of BDI-II scores. Because of the likely bimodal distribution, it would be more appropriate to look at links between depression and resting state functional connectivity between the depressed sample and healthy control sample separately or just amongst the depressed sample

Results

  1. It would be important to show scatter plots of BDI-Resting State Connectivity associations, childhood trauma-resting state functional connectivity associations, and a figure showing childhood trauma as a moderator of BDI-resting state connectivity associations. These scatter plots likely would highlight the bimodal distribution of BDI-II scores.
  2. It seems that in the text, stats are included along with degrees of freedom information for childhood abuse-resting state functional connectivity associations, childhood neglect-resting state functional connectivity associations, and childhood trauma as a moderator between BDI-II scores and resting state functional connectivity, however, stats are not included in the text for BDI-II – resting state functional connectivity associations. These stats need to be added to the text.

Author Response

Point 1: The authors mentioned in the introduction starting on line 101:” Since different childhood trauma subtypes have differential effects on 102 regional brain structures, activity, and connectivity [34], we decided to analyse abuse and 103 neglect separately by focusing on emotional and physical abuse and emotional and physical neglect, whereas we did not analyse the effects of sexual abuse.” It would be helpful to expand on these prior findings…what do we know about the different neural underpinnings of different trauma types? This will help make it clearer how you derived your hypotheses.

Response 1: In their systematic review, Cassiers et al. (2018) conclude that sexual abuse is associated with structural deficits in the reward circuitry and genitosensory cortex, as well as amygdalar hyperreactivity during sad autobiographic memory recall. Emotional maltreatment correlated with abnormalities in fronto-limbic socioemotional networks, whereas in neglected individuals white matter integrity and connectivity were disturbed in several brain networks involved in a variety of functions. Changes to the manuscript have been done according to the reviewer‘s suggestion in the introduction section, to better expand on Cassiers et al. (2018). Furthermore, based on suggestions by other reviewers, we included some more theoretical grounding, introducing the dimensional model of adversity and psychopathology (DAMP; see lines 102 – 128 in the introduction section), hoping that this also explains why we decided to apply a dimensional approach.

Lines 102-128: Different approaches have been applied when analyzing the effects of ELA. The cumulative risk model accounts for the observation that different types of adversities show high co-occurrence, focusing on the number of distinct adverse experiences rather than severity and type of experience [34, 35]. Although this approach has proven useful in establishing a dose-response relationship between ELA exposure and negative health outcomes, it implies that different forms of adversity share the same underlying mechanisms, therefore showing significant limitations when used to identify mechanisms linking ELA with developmental and health outcomes [35]. Recently, an alternative model has been proposed, namely the dimensional model of adversity and psychopathology (DAMP) [36, 37])[37][35] In line with the assumptions of the DAMP, it has recently been stated that different childhood trauma subtypes go along with specific neural underpinnings, with emotional maltreatment being linked especially to abnormalities in fronto-limbic socioemotional networks, neglect showing associations with abnormalities of white matter integrity and FC in various brain networks (especially an insula-based network) and sexual abuse more strongly correlating with structural deficits (especially in the reward circuit and genitosensory cortex), as well as amygdalar reactivity to certain tasks [38]. In line with the assumptions of the DAMP and based on previous findings regarding specific effects of ELA subtypes on brain measurements, we decided to analyze abuse and neglect separately by focusing on emotional and physical abuse and emotional and physical neglect.

Point 2: Also, there is a type in the above sentences “analyse” should be “analyze”. I think “analyse” is the UK spelling.

Response 2: We apologize for this oversight and corrected it accordingly.

Point 3: In the introduction, it was stated that the effects of sexual abuse was not analyzed. Why is that? No explanation is provided.

Response 3: Maltreatment and neglect are associated with abnormal connectivity in various brain networks, whereas sexual abuse more strongly correlates with structural deficits (escpecially in the reward circuit and genitosensory cortex) and amygdalar reactivity to certain tasks (Cassiers et al., 2018). Therefore, we decided not to analyze sexual abuse, since the hypothesized influences of sexual abuse can not be shown in our analyses due to lack of appropriate outcome variables. Furthermore, we point to prior research by Carpenter et al. (2009, see lines 128-131) showing that sexual abuse shows distinct stress reactivity patterns when compared to other forms of ELA. We now include these findings in the introduction section, thereby making our rationale for not analyzing effects of sexual abuse more clear to the readers.

Lines 128-131.: Due to previous reports of sexual abuse showing distinct patterns of stress reactivity when compared with other forms of ELA [39] and its specific influence on measurements of brain structure and reactivity [38], we decided to exclude this subscale from analyses.

Point 4: It sounds like in the methods section that the authors looked at associations between BDI-II scores and functional connectivity across healthy controls and those with major depressive disorder? If so, this is likely highly problematic, given that you likely have a bimodal distribution of BDI-II scores. Because of the likely bimodal distribution, it would be more appropriate to look at links between depression and resting state functional connectivity between the depressed sample and healthy control sample separately or just amongst the depressed sample

Response 4: We would like to thank the reviewer for raising this very important point and added a paragraph to better address the limitation of our approach in the limitations section (see lines 500-503). This being said, we believe that our dimensional approach is supported by the broad distribution of BDI-II scores in the sample, i.e. a significant proportion of depressed individuals show subclinical or rather low scores in self-reported depression severity, while some of the healthy individuals show some depressive symptoms (scatterplots are now provided in the Supplementary Material, according to Point 5). Furthermore, the dimensional approach is in line with the DAMP we mentioned in Response 1.

Lines 500-503.: However, this dimensional approach also implied that we combined two groups in which symptom severity was unequally distributed at the danger of inflating certain associations of symptoms and FC abnormality.

Point 5: It would be important to show scatter plots of BDI-Resting State Connectivity associations, childhood trauma-resting state functional connectivity associations, and a figure showing childhood trauma as a moderator of BDI-resting state connectivity associations. These scatter plots likely would highlight the bimodal distribution of BDI-II scores.

Response 5: Following the reviewers‘ suggestion, we now included scatter plots of BDI-Resting State Connectivity associations, childhood trauma-resting state functional connectivity associations, and a figure showing childhood trauma as a moderator of BDI-resting state connectivity associations in the supplementary material.

Point 6: It seems that in the text, stats are included along with degrees of freedom information for childhood abuse-resting state functional connectivity associations, childhood neglect-resting state functional connectivity associations, and childhood trauma as a moderator between BDI-II scores and resting state functional connectivity, however, stats are not included in the text for BDI-II – resting state functional connectivity associations. These stats need to be added to the text.

Response 6: Stats for BDI-II – resting state functional connectivity associations (degrees of freedom, T-value, p-FDR value) are provided the same way as they are provided for the other associations, namely in table 2 (lines 286 – 292). For CTQ_abuse – resting state functional connectivity associations, we described the stats in the text (because there was only one pair of ROIs showing a significant association), with the same procedure being applied for the moderation (for the same reason), whereas for BDI-resting state functional connectivity associations and neglect-resting state functional connectivity associations we show the stats in tables (since there are more connections that have to be reported and therefore this provides a better overwiew from our point of view).

Lines 286-292:

Table 2. Significant associations between BDI-II and ROI-to-ROI FC, sorted by within- and between-network FC.

Pair of ROIs

Dir

T(75)

p-FDR

Within-network FC

Salience network

rPFC (left) – rPFC (right)

pos

2.58

.04

rPFC (left) - ACC

pos

2.49

.04

rPFC (left) – SMG (right)

pos

2.45

.04

Between-network FC

Salience network – Default mode network

rPFC (left) – LP (left)

neg

-3.23

.01

rPFC (left) – LP (right)

neg

-2.29

.04

Salience network – Central executive network

rPFC (left) – LPFC (left)

neg

-3.91

.003

rPFC (left) – PPC (left)

neg

-3.12

.01

rPFC (left) – PPC (right)

neg

-2.37

.04

Notes. ROI = region of interest, Dir = Direction, FC = Functional connectivity, rPFC = rostral prefrontal cortex, ACC = anterior cingulate cortex, SMG = supramarginal gyrus, LP = lateral parietal, LPFC = lateral prefrontal cortex, PPC = posterior parietal cortex, pos = positive, neg = negative, unc = uncorrected, FDR = false discovery rate.

 Additional comment:

The reviewer asked for a categorical analysis, maybe as part of supplementary analyses. Following the reviewer’s suggestion, we tried to build groups based on the approach described in the paper the reviewer cited (Fan et al., (2020), Childhood trauma is associated with elevated anhedonia and altered core reward circuitry in major depression patients and healthy controls. Human Brain Mapping, 1-12.). We had 20 participants reporting traumatization on the moderate-to-severe range on any of the subscales (according to the cut-off-scores in Fan et al., 2020) and 63 participants without moderate-to-severe traumatization. If – according to our procedure when performing the dimensional analyses - we would exclude the participants reporting sexual abuse (n = 3), the groups would be even more imbalanced. We therefore do not expect the analyses to be informative, given the disbalanced group numbers. Furthermore, the theoretical framework of the current work is in line with the DAMP (see Responses above and added paragraphs in the manuscript), therefore a categorical analysis would be difficult to be embedded in our work. From our point of view, such analyses do not enter the scope of the current work. We thank the reviewer very much for the input, but would suggest future studies to perform categorical analyses, as such analyses would rather show general trauma effects (since the effects cannot be clearly linked to trauma subtypes) and this would represent a different approach to the matter.

Reviewer 2 Report

The authors have done good work describing the study design. Taken together, they provide evidence for FC patterns associated with the severity of depression symptoms which is important to better understand the mechanisms underlying this disease. However, there were some areas not clearly presented and few gaps in the information. Please see the following comments: 

  1. Lateralization of the insula, accompanied by sex differences, has been previously described (see Duerden et al. 2013 as an example, DOI: 10.1016/j.neuroimage.2013.04.014). In the current study authors only show association for right insula. The left insula was not mentioned throughout the study which makes it impossible to compare and to better appreciate the changes presented for the right side. The discussion and results should include the contribution of sex differences since this study group has disproportionately more women (32) than men (10). 
  2. As mentioned in the methods, the authors combined subscales but also showing these individually will be informative when comparing to other studies. 
  3. The study aims to examined associations between FC and depressive symptoms severity. However, the overall symptoms assessed in the BDI-II test were not described. Providing detailed information about the depressive symptoms and how the severity is influenced by gender or age will be important to support the conclusions.
  4. Table 3 and Figure 2 correspond to associations with CTQ neglect. But these were not presented for CTQ abuse. Please include these data.  
  5. One of the limitations is the age cohort. Splitting data into two age subgroups (example: young adults, 18 to 40, and older adults 41 to 68 years) will be informative to understand more about age-onset and FC patterns. 

Author Response

Response to Reviewer 2 Comments

Point 1: Lateralization of the insula, accompanied by sex differences, has been previously described (see Duerden et al. 2013 as an example, DOI: 10.1016/j.neuroimage.2013.04.014). In the current study authors only show association for right insula. The left insula was not mentioned throughout the study which makes it impossible to compare and to better appreciate the changes presented for the right side. The discussion and results should include the contribution of sex differences since this study group has disproportionately more women (32) than men (10). 

Response 1: We thank the reviewer for raising this very important point. It is true that in our study women are more strongly represented than men (63 women and 20 men). We controlled for sex differences in the analyses, so these should not influence the reported effects. When analyzing the influence of sex – and controlling for all the other mentioned variables – we did not find any significant contribution of sex to FC between any pair of ROIs (all p-FDRs > 0.05). As for the left insula, results are not mentioned because BDI, CTQ abuse, CTQ neglect, and the interaction terms (CTQ*abuse and CTQ*neglect) did not show a significant association with FC between the left insula and any other examined ROI (all p-FDRs > 0.05). Following the reviewers’ suggestion, we mention this point as a possible limitation to our study and included the mentioned literature (see lines 510-515).

Lines 510-515: Second, in our sample women are more strongly represented than men, an aspect that should be noted in the light of research pointing to sex differences in resting state FC [67, 68]. Although we controlled for sex differences in our analyses, the observed results might be more representative of influences of severity of depressive symptoms and childhood trauma in female rather than male subjects.

Point 2: As mentioned in the methods, the authors combined subscales but also showing these individually will be informative when comparing to other studies. 

Response 2: Following the reviewer’s suggestion, we added the subscales to table 1 in the results section (lines 270-275).

Lines 270-275:

Table 1. Sociodemographic and psychometric characteristics of the sample.

Variables

Healthy control group (n = 42)

Depressive patients (n = 41)

Age

M = 30.74 (SD = 11.22)

M = 28.2 (SD = 10.06)

Gender

       Female

       Male

n = 32

n = 10

n = 31

n = 10

BDI-II

M = 2.43 (SD = 2.92)

M = 27.1 (SD = 9.04)

CTQ: Neglect

M = 14.55 (SD = 5.31)

M = 16.73 (SD = 6.27)

CTQ: Abuse

M = 12.38 (SD = 3.65)

M = 15.27 (SD = 6.34)

CTQ: SA

M = 5.12 (SD = 0.55)

M = 5.44 (SD = 1.23)

CTQ: EN

M = 8.12 (SD = 3.83)

M = 10.22 (SD = 4.53)

CTQ: PN

M = 6.43 (SD = 2.06)

M = 6.51 (SD = 2.27)

CTQ: EA

M = 7 (SD = 2.66)

M = 9 (SD = 3.80)

CTQ: PA

M = 5.38 (SD = 1.58)

M = 6.27 (SD = 3.05)

Notes. BDI-II = Beck Depression Inventory-II. CTQ = Childhood Trauma Questionnaire. SA = Sexual Abuse. EN = Emotional Neglect. PN = Physical Neglect. EA = Emotional Abuse. PA = Physical Abuse.

Point 3: The study aims to examined associations between FC and depressive symptoms severity. However, the overall symptoms assessed in the BDI-II test were not described. Providing detailed information about the depressive symptoms and how the severity is influenced by gender or age will be important to support the conclusions.

Response 3: We would like to thank the reviewer for the suggestion. To our knowledge, there is no empirical framework for testing gender or age effects on severity of depressive symptoms (see Wang & Gorenstein, 2013, for a review article regarding the BDI-II; this reference has been added to our manuscript together with a description of the instrument, see lines 194-195). Furthermore, we included age and gender as covariates to control for their effects in the analyses and are therefore confident that our results should not be influenced by age or gender.

Lines 194-195: To assess the severity of depressive symptoms we used the Beck Depression Inventory II, a self-report questionnaire assessing general depression (BDI-II [43, 44]).

Point 4: Table 3 and Figure 2 correspond to associations with CTQ neglect. But these were not presented for CTQ abuse. Please include these data.  

Response 4: The associations for CTQ abuse were not presented in a table because only one pair of ROIs (namely the right rPFC and right Insula) showed significant associations with CTQ abuse. The stats for the association are mentioned in the text (lines 300-301). For CTQ_abuse – resting state functional connectivity associations, we described the stats in the text because there was only one pair of ROIs showing a significant association, with the same procedure being applied for the moderation (for the same reason), whereas for BDI-resting state functional connectivity associations and neglect-resting state functional connectivity associations we show the stats in tables (since there are more connections that have to be reported and therefore this provides a better overview from our point of view).   

Lines 300-301: CTQ abuse was positively associated with FC within the SN, namely with FC of the right insula with the right rPFC (T(75) = 3.40, p-FDR = .02).

Point 5: One of the limitations is the age cohort. Splitting data into two age subgroups (example: young adults, 18 to 40, and older adults 41 to 68 years) will be informative to understand more about age-onset and FC patterns. 

Response 5: We thank the reviewer for raising this important point and fully agree that age per se might have affected the results. However, subjects in our sample are rather young and therefore, older adults are not adequately represented. The suggested split of age groups would accordingly have resulted in considerably disbalanced numbers of subjects per group, i.e. N= 71 subjects in the young and 12 subjects in the old group, when splitting the sample at age 40. We, however, included the young age of the sample in the limitations and suggested for future studies to test the associations between depression and FC and between childhood trauma and FC in older samples (lines 526-544). Furthermore, we included age as a control variable in our analyses, and are therefore confident that the reported effects should not be influenced by age.  

Lines 526-544: Fourth, our sample is relatively young (overall mean of 29.47 years). While many depressive disorders have their onset in late adolescence and early adulthood, they most often take a recurrent and chronic treatment course [70]. Our sample is likely not representative of these recurrent and chronic treatment courses. Also, while it is well documented that ELA influences mental health outcomes across the lifespan [71,72], the surely complex association of time and neurobiological correlates of ELA remains to be clarified – a matter that we were not able to address in our study. Thus, our findings need to be replicated in samples older in age or ideally in a longitudinal study design.

Reviewer 3 Report

This is an original investigation on the effects of early life adversity and  current depression on resting state functional connectivity (FC) of several relevant large-scale networks in 83 depressed and healthy control adults. Results showed that depressive symptoms were linked to increased FC within the salience network (SN) and decreased FC of the SN with the DMN and CEN. Also, differential associations of childhood abuse and neglect were found in different directions and networks, mostly consistent with previous findings. Results are discussed in the context of related research and reference to relevant studies has generally been made. The topic of the study is very relevant for the journal and actual, the paper is very well-written overall and the authors have used statistical analyses competently. However, the paper needs some significant improvement across the sections to meet the standards for publication in the Journal, particularly resulting in a higher understanding of and better formulated theoretical grounding and a more complete and integrated interpretation of their findings within the relevant theories. I have the following more specific suggestions and comments on the various sections of the paper:

  1. Can the authors add some more theoretical grounding in their Introduction with regard to the general as well as the specific associations that can supposedly link specific forms of early adversity to later depression? Which of the proposed models have the authors made most use of to formulate their hypotheses since new theories have lately emerged as potential alternative explanations to earlier models (e.g., the Dimensional Model of Adversity and Psychopathology (DMAP; see, e.g, McLaughlin, Sheridan & Lambert, 2014; Sheridan & McLaughlin, 2014) versus the Cummulative Risk model)? This could possibly be linked to the general approach that the authors have used when analyzing their data (dimensional rather than a group-based approach, as mentioned in lines 402-404).
  2. Can the authors include information on the distributions of CTQ and depression scores and whether data transformation methods were needed or not?
  3. A significant negative association was reported between the interaction term BDI-II*abuse and FC within the SN (lines 269-270) with the authors suggesting significant associations in opposing directions between depression scores and FC in the high vs. low abuse conditions (lines 271-273). Can the authors present the statistical coefficients showing these specific associations?
  4. The Discussion section where the authors interpret the presence of a moderation effect for abuse but not neglect (section 4.3 and particularly lines 398-400) should be treated more extensively and preferably be linked to the theoretical grounding previously identified, as suggested above. These differential associations reported for abuse versus neglect with regard to neuroimaging findings in depression represent an excellent opportunity for the authors to push forward the current knowledge in this very narow but important developmental psychopathology area and use their empirical findings to draw important conclusions more firmly and that could prove beneficial not only for researchers but also for therapists.

Minor suggestions:

  1. ”20 of these subjects” (line 138) should be changed to ”Twenty of these subjects”.
  2. The authors should mention the statistical program(s) that they used to analyze their data.

Author Response

Response to Reviewer 3 Comments

Point 1: Can the authors add some more theoretical grounding in their Introduction with regard to the general as well as the specific associations that can supposedly link specific forms of early adversity to later depression? Which of the proposed models have the authors made most use of to formulate their hypotheses since new theories have lately emerged as potential alternative explanations to earlier models (e.g., the Dimensional Model of Adversity and Psychopathology (DMAP; see, e.g, McLaughlin, Sheridan & Lambert, 2014; Sheridan & McLaughlin, 2014) versus the Cummulative Risk model)? This could possibly be linked to the general approach that the authors have used when analyzing their data (dimensional rather than a group-based approach, as mentioned in lines 402-404).

Response 1: We thank the reviewer for this important contribution and for the literature suggestions. We added a paragraph describing the DAMP, since our hypotheses and analyses approach are strongly in line with the assumptions of the model (see lines 102-128 in the introduction section). Furthermore, we included the model in our discussion, as described in Response 4.

Lines 102-128: Different approaches have been applied when analyzing the effects of ELA. The cumulative risk model accounts for the observation that different types of adversities show high co-occurrence, focusing on the number of distinct adverse experiences rather than severity and type of experience [34, 35]. Although this approach has proven useful in establishing a dose-response relationship between ELA exposure and negative health outcomes, it implies that different forms of adversity share the same underlying mechanisms, therefore showing significant limitations when used to identify mechanisms linking ELA with developmental and health outcomes [35]. Recently, an alternative model has been proposed, namely the dimensional model of adversity and psychopathology (DAMP) [36, 37])[37][35] In line with the assumptions of the DAMP, it has recently been stated that different childhood trauma subtypes go along with specific neural underpinnings, with emotional maltreatment being linked especially to abnormalities in fronto-limbic socioemotional networks, neglect showing associations with abnormalities of white matter integrity and FC in various brain networks (especially an insula-based network) and sexual abuse more strongly correlating with structural deficits (especially in the reward circuit and genitosensory cortex), as well as amygdalar reactivity to certain tasks [38]. In line with the assumptions of the DAMP and based on previous findings regarding specific effects of ELA subtypes on brain measurements, we decided to analyze abuse and neglect separately by focusing on emotional and physical abuse and emotional and physical neglect.

Point 2: Can the authors include information on the distributions of CTQ and depression scores and whether data transformation methods were needed or not?

Response 2: Following the reviewer’s suggestion, information about the distributions of CTQ and depression scores have been added to the text in the results section (see lines 264-266). The limitations of our dimensional approach are discussed more thoroughly in the limitations section (see lines 500-503). The BDI-II, CTQ-abuse and CTQ-neglect scores have been mean centered to form the interaction terms of BDI*abuse and BDI*neglect, as described in lines 247-250 in the statistical analysis section.

Lines 247-250: We used the following predictors: BDI-II score, CTQ abuse score, CTQ neglect score, BDI-II*abuse, BDI-II*neglect, age, and gender. To form the interaction terms of BDI-II with CTQ abuse and BDI-II with CTQ neglect, the BDI-II score as well as the CTQ abuse and CTQ neglect scores were mean centered and then the product between BDI-II and CTQ abuse scores and between BDI-II and CTQ neglect scores was calculated.

Lines 264-266: Here, we report associations of ROI-to-ROI FC with BDI-II (M = 14.61, SD = 14.08, range = 0-47), CTQ abuse (M = 13.81, SD = 5.33, range = 9-46) and CTQ neglect (M = 15.63, SD = 5.87, Range = 10-36) scores.

Lines 500-503: However, this dimensional approach also implied that we combined two groups in which symptom severity was unequally distributed at the danger of inflating certain associations of symptoms and FC abnormality.

Point 3: A significant negative association was reported between the interaction term BDI-II*abuse and FC within the SN (lines 269-270) with the authors suggesting significant associations in opposing directions between depression scores and FC in the high vs. low abuse conditions (lines 271-273). Can the authors present the statistical coefficients showing these specific associations?

Response 3: We would like to thank the reviewer for pointing out our possibly misleading formulation, that may imply we calculated the coefficients of the single associations. The statistical coefficients regarding the interaction term are reported in lines 319 – 325. As for the specific associations, we interpreted them based on visual inspection. We now include a figure showing the moderation effect in the supplementary materials and changed our formulation in the results section (lines 319-325).

Lines 320-326: The interaction term BDI-II*abuse was negatively associated with FC within the SN, namely with FC of the right insula with the right rPFC ((T(75) = -3.24, p-FDR = .02). For participants with higher scores of CTQ abuse, BDI-II was associated with decreased FC of the right insula with the right rPFC. These associations were inverse for participants with lower scores of CTQ abuse (see figure S4 in the supplementary materials). We found no significant association of FC with the interaction between BDI-II and CTQ neglect (BDI*neglect; all p-FDRs > .05).

Point 4: The Discussion section where the authors interpret the presence of a moderation effect for abuse but not neglect (section 4.3 and particularly lines 398-400) should be treated more extensively and preferably be linked to the theoretical grounding previously identified, as suggested above. These differential associations reported for abuse versus neglect with regard to neuroimaging findings in depression represent an excellent opportunity for the authors to push forward the current knowledge in this very narow but important developmental psychopathology area and use their empirical findings to draw important conclusions more firmly and that could prove beneficial not only for researchers but also for therapists.

Response 4: We thank the reviewer for this very important contribution. The DAMP has been added in the introduction section, as stated in Response 1. In addition, we included the model for the discussion of our results (lines 446-449, lines 476-495). We hope the more extensive discussion meets the reviewer’s suggestion.

Lines 446-449: Taken together, our findings are in line with the notion that different childhood trauma subtypes go along with specific neural underpinnings [38] and support the DAMP’s suggestion to analyze the effects of threat (here defined as abuse) and deprivation (here defined as neglect) separately, when controlling for either one, respectively [62].

Lines 476-495: Finally, our observation of a moderation effect regarding childhood abuse, but not neglect, is in line with previous reports suggesting that unique developmental mechanisms link different types of ELA with psychopathology [62, 65]. According to the DAMP, experiences of threat (here defined as childhood abuse) are supposed to particularly influence brain circuits involved in emotional processing, increasing risk for psychopathology by leading to difficulties with emotion regulation [62]. Our observation of a moderating effect of abuse on the association between severity of depressive symptoms and FC within the SN is in line with this assumption. The DAMP assumes that deprivation (here defined as neglect) influences psychopathology by altering the development of complex cognition and associated neural substrates (e.g. the CEN) [62], an assumption which we have not been able to prove in our study. Recently, it has been reported that deprivation was associated with risk for externalizing problems via effects on verbal abilities [66]. We therefore suggest for future research to analyze the language network in association with threat and deprivation. A better understanding of the mechanisms through which different forms of ELA link to psychopathology would be highly relevant for treatment interventions, as it could be suggested that depressed individuals with experiences of abuse might profit more strongly form therapeutic interventions focusing on emotion regulation abilities, whereas individuals with a history of neglect might profit more from interventions targeting cognitive and verbal abilities. These suggestions need to be addressed by future research.

Minor suggestions:

Point 5: ”20 of these subjects” (line 138) should be changed to ”Twenty of these subjects”.

Response 5: We thank the reviewer for outlining this point, it was revised as requested.

Point 6: The authors should mention the statistical program(s) that they used to analyze their data.

Response 6: To analyze fMRI-data, the toolbox CONN was used (see line 216). Statistical analyses were also performed with CONN, we added this information in the paragraph 2.2.3 statistical analysis (line 245).

Line 216: Resting state fMRI data was analyzed using the toolbox CONN [46].

Line 245: Analyses were performed using the toolbox CONN [46].
